# The effect of listening to music on human transcriptome

Chakravarthi Kanduri[1], Pirre Raijas[2], Minna Ahvenainen[1], Anju K. Philips[1], Liisa Ukkola-Vuoti[1], Harri Lähdesmäki[3] and Irma Järvelä[1]

[1] Department of Medical Genetics, University of Helsinki, Finland
[2] DocMus Department, University of the Arts Helsinki, Helsinki, Finland
[3] Department of Information and Computer Science, Aalto University, AALTO, Finland

Corresponding author
Irma Järvelä,
irma.jarvela@helsinki.fi

## ABSTRACT

Although brain imaging studies have demonstrated that listening to music alters human brain structure and function, the molecular mechanisms mediating those effects remain unknown. With the advent of genomics and bioinformatics approaches, these effects of music can now be studied in a more detailed fashion. To verify whether listening to classical music has any effect on human transcriptome, we performed genome-wide transcriptional profiling from the peripheral blood of participants after listening to classical music ($n = 48$), and after a control study without music exposure ($n = 15$). As musical experience is known to influence the responses to music, we compared the transcriptional responses of musically experienced and inexperienced participants separately with those of the controls. Comparisons were made based on two subphenotypes of musical experience: musical aptitude and music education. In musically experiencd participants, we observed the differential expression of 45 genes (27 up- and 18 down-regulated) and 97 genes (75 up- and 22 down-regulated) respectively based on subphenotype comparisons (rank product non-parametric statistics, pfp 0.05, >1.2-fold change over time across conditions). Gene ontological overrepresentation analysis (hypergeometric test, FDR < 0.05) revealed that the up-regulated genes are primarily known to be involved in the secretion and transport of dopamine, neuron projection, protein sumoylation, long-term potentiation and dephosphorylation. Down-regulated genes are known to be involved in ATP synthase-coupled proton transport, cytolysis, and positive regulation of caspase, peptidase and endopeptidase activities. One of the most up-regulated genes, *alpha-synuclein* (*SNCA*), is located in the best linkage region of musical aptitude on chromosome 4q22.1 and is regulated by *GATA2*, which is known to be associated with musical aptitude. Several genes reported to regulate song perception and production in songbirds displayed altered activities, suggesting a possible evolutionary conservation of sound perception between species. We observed no significant findings in musically inexperienced participants.

## INTRODUCTION

Listening to music is common in all societies. A plethora of neurophysiological studies have demonstrated that listening to and/or playing music has multiple measurable effects on human brain structure and function (*Elbert et al., 1995*; *Blood & Zatorre, 2001*; *Sutoo & Akiyama, 2004*; *Koelsch, 2011*; *Salimpoor et al., 2011*; *Salimpoor et al., 2013*; *Herholz & Zatorre, 2012*; *Chanda & Levitin, 2013*). In investigations using positron emission tomography (PET), music listening has been reported to cause physiological changes in cerebral blood flow, cardiovascular and muscle function, and enhanced dopamine secretion in the human brain (*Sutoo & Akiyama, 2004*; *Angelucci et al., 2007*; *Salimpoor et al., 2011*). Music has been demonstrated to regulate emotions and evoke pleasure, primarily through its action on the brain's reward centres like the limbic and mesolimbic structures including the nucleus accumbens, hypothalamus, subcallosal cingulate gyrus, pre-frontal anterior cingulate, and hippocampus (*Brown, Martinez & Parsons, 2004*; *Koelsch, 2011*; *Koelsch, 2014*; *Salimpoor et al., 2011*; *Salimpoor et al., 2013*). Music has also been used as a therapeutic tool in clinical settings (*AMTA, 2014*; *Conrad, 2010*; *Holmes, 2012*). However, the molecular mechanisms and biological pathways mediating the effects of music remain unknown.

Genomics and bioinformatics offer methods (*Lander, 2011*) to explore the biology and evolution of music and sounds at the molecular level. To date, few genome-wide scans have been performed in musical traits in humans (*Asher et al., 2008*; *Pulli et al., 2008*; *Park et al., 2012*; *Oikkonen et al., 2014*). Genome-wide expression analysis can be applied to study human traits in an unbiased, hypothesis-free fashion based on their molecular properties, rather than anatomic regions. Here, we have utilized a combination of genomic and bioinformatic methods to analyze the effect of classical music on the peripheral whole blood transcriptome. Peripheral blood was used, as brain samples are inaccessible in humans.

## MATERIALS AND METHODS

### Ethics statement

The Ethics Committee of Helsinki University Central Hospital approved this study. Written informed consent was obtained from all the participants.

### Participants and phenotypes

A total of 48 individuals (aged 18–73; mean 42.5) participated in the study (Table S1). All the 48 participants were characterized for musical aptitude and music education. Musical aptitude was measured using three tests: the auditory structuring ability test (Karma music test) (*Karma, 2007*), and the Seashore tests for pitch and for time (*Seashore, Lewis & Saetveit, 1960*). Details of the tests have been described in *Oikkonen et al. (2014)*. A combined score of the three tests (COMB score; range 0–150) was used to define musical aptitude. COMB scores were classified as either high or low based on the upper and lower quartiles of score distribution. Data on music education were collected using a questionnaire. The self-reported levels of music education (referred to as *edu classes* 1–4), received through studies or degrees from music schools, institutes, or universities, were

**Peer**J ______________________________________

**Table 1  Phenotype characteristics of the sample set (total _n_ = 48).**

| Phenotype | _n_ [*] |
|---|---|
| _Edu classes 1–2_ | 19 |
| _Edu classes 3–4_ | 29 |
| _Low COMB scores_ | 12 |
| _High COMB scores_ | 12 |
| Male | 22 |
| Female | 26 |

**Notes.**

[*] _n_ represents the number of participants within each category.

classified as follows: 1 represents no music education; 2 represents music education of less than two years; 3 represents music education of more than ten years; and 4 represents a professional musician. The approximate time of systematic music education and training was 21.42 years on average, in participants of _edu classes 3–4_. Table 1 shows the sample's phenotype characteristics.

## Exposure to music

To our knowledge, no previous studies have systematically studied the effect of listening to music on genome-wide transcriptional profiles of humans. We have previously shown that music-listening habits vary a lot among listeners (_Ukkola-Vuoti et al., 2011_). To start with, we chose to study the effect of classical music, Wolfgang Amadeus Mozart's Violin Concerto No. 3 in G major, K.216 because it is relatively familiar in the western culture. As the human brain perceives complex sounds in a millisecond-level time frame (_Wang et al., 2009_; _Kayser, Logothetis & Panzeri, 2010_), we expected that the 20 min-listening session (duration that the concerto lasts) will induce an effect of music on human transcriptome. In studies on the effect of pain in humans, very short durations of pain induction (8 to 150 s) have been used (_Hubbard et al., 2011_). The participants were unaware of the type of music that was intended for the listening session. Peripheral blood samples were collected from all the 48 participants just before and after the listening session. From here on, participants who listened to music are referred to as _listeners_ throughout the text.

## Control study

The same 48 participants were invited to a control study. Of these, 15 participants could attend. The participants were advised to avoid listening to music and hard exercise the day before the control study. The control study was performed in a 'music-free' environment, where the participants had an opportunity to converse, read a magazine, or take a walk outside (no exercise) during the session. Peripheral blood samples were collected from the participants just before and after 20 min in the control session (the same duration as in the listening session). From here on, participants of the control session are referred to as _controls_ throughout the text.

## Genome-wide expression profiling

For this procedure, $2 \times 2.5$ ml samples of peripheral blood were drawn into PAXgene blood RNA tubes (PreAnalytiX GmbH, Hombrechtikon, Switzerland) as per the kit instructions, in both of the sessions. Total RNA was isolated using the PAXgene blood miRNA Kit (PreAnalytiX GmbH, Hombrechtikon, Switzerland) as per the kit manual. Purified RNA samples were measured for concentration and purity on the NanoDrop 1000 v.3.7 (Thermo Fisher Scientific, Waltham, Massachusetts, USA). Globin mRNA was depleted from our samples using Ambion's Human GLOBINclear$^{TM}$ kit (Applied Biosystems, Carlsbad, California, USA) as per the kit insert. The samples were measured on the NanoDrop 1000 to determine the sample concentration and purity and for integrity on the 2100 Bioanalyzer (Agilent Technologies, Waldbronn, Germany) before being diluted to 50 ng/µl using RNase-free water. A total of 2 µg of RNA was assayed on the Illumina HumanHT-12 v4 bead array (Illumina Inc., San Diego, California, USA), which targets more than 47,000 probes. The gene expression profiling assays for the listening and control sessions were conducted in two separate batches. To account for the batch effect corrections, six samples from the listening session were assayed together with the control session samples. Intensity data were exported through Bead Studio software. The data reported in this article have been deposited in the Gene Expression Omnibus database, www.ncbi.nlm.nih.gov/geo (accession no. GSE48624).

## Data preprocessing

The Lumi package was used to read and preprocess the signal intensity data. Specifically, pre-processing included background correction, variance stabilizing transformation, and quantile normalization. Data from both the listening ($n = 48$) and control ($n = 15$) sessions were normalized separately. Five samples from the control session were excluded from further analyses owing to data quality. In addition, we used the ComBat method (*Johnson, Li & Rabinovic, 2007*) to adjust for batch effects and determine if this correction affected the pre-post fold-changes across conditions. However, we did not find significant differences between the fold-changes of corrected and uncorrected data over time across conditions. Therefore, we chose to proceed with the uncorrected data owing to the strengths of our analysis methods as described below. After normalization, duplicate and un-annotated probes were excluded using the genefilter package (R package version 1.40.0). Before extracting the expression values from the normalized data, Illumina's detection $p$-values (threshold: 0.01) were used to filter out probes with low intensities corresponding to the background signal. Finally, only those probes that were expressed in at least half of all of the arrays (listening and control sessions) were chosen for the study.

## Differential expression analysis

The choice of an appropriate statistical test for the identification of differentially expressed genes depends upon several aspects of the data including the underlying distribution, homogeneity, and the sample size. As the statistical tests for normality are known to be sensitive to sample size, we used a normal Q-Q plot to get a glimpse of the distribution of

the data. For this, we randomly visualized the distribution of transcriptional responses of control samples ($n = 10$) for several transcripts (Fig. S1) using normal Q-Q plots. We observed that the data appeared to deviate from normality in several instances. As the central limit theorem does not always hold true for small sample sizes, a cautious approach here would be to employ a non-parametric test (better being safe than sorry). Non-parametric tests are deemed to be appropriate analysis tools when the distribution of data is difficult to characterize, because they make less stringent distributional assumptions. Therefore, we chose to use the rank product (*Breitling et al., 2004*) non-parametric test statistic, which is relatively powerful especially for small sample sizes, and when the data is heterogeneous and does not meet normality. Rank product-based methods outperformed several other methods including empirical Bayes statistic (limma) and SAM, when the sample size is small and when the data is non-homogeneous (*Jeffery, Higgins & Culhane, 2006*). Also, a comparison of eight gene ranking methods using Microarray Quality Control datasets (golden-standard) has demonstrated the high sensitivity and specificity of the rank product method (*Kadota & Shimizu, 2011*).

To identify the differentially expressed genes, we compared the magnitude of pre-post changes in gene expression across conditions using the rank product method implemented in the RankProd Bioconductor package. Based on the estimated percentage of false predictions (pfp), RankProd employs a non-parametric statistic to identify genes that are consistently ranked high among the most up- or down-regulated genes in replicate experiments. Instead of analyzing the actual expression value, this method utilizes the ranks of genes in each sample. The strength of rank product method allows us to compare and combine the datasets of the listening and control studies. After the identification of differentially expressed genes using a pfp of 0.05 in RankProd, we selected only those genes that exceeded an effect-size cut-off ($>1.2$-fold change over time across conditions, and at least a pre-post change of 10% in gene expression in the listening session). Here, we would like to point out a couple of aspects of these selection criteria. First, pfp employed by RankProd is equivalent to the standard false discovery rate (FDR). Second, there exists a widespread misconception that only two-fold changes are significant (*Hoheisel, 2006*) and that false notion is based on the very initial publications of microarray studies, which used a two-fold change criteria for a particular group of experiments because of biological relevance. Fold-change thresholds are completely arbitrary and in the majority of the cases they depend upon the underlying biological question. For example, studies that investigated the effect of gene-environment interactions (socio-environmental effect (*Cole et al., 2007*), yogic meditation effect (*Black et al., 2012*)) used unorthodox fold-change thresholds.

We further performed successive functional annotation analyses using GeneTrail (http://genetrail.bioinf.uni-sb.de/) and IPA (Ingenuity® Systems, www.ingenuity.com). The gene ontological overrepresentation analysis in GeneTrail uses a hypergeometric distribution test along with a conservative multiple testing correction method (FDR $< 0.05$) to assess whether genes belonging to certain functional categories are enriched in the dataset. In addition to the standard gene ontology analyses, we performed upstream transcription regulator analysis, which essentially predicts all the upstream transcription

regulators (transcription factors, receptors, cytokines, microRNA, and kinases) that could have possibly mediated the observed differential expression. Based on the overlap between known targets of a transcription regulator and the set of differentially expressed genes, an overlap $p$-value is computed using Fisher's exact test ($p < 0.01$). Further, the activation states of the predicted transcription regulators are also inferred using an activation Z-score, which is based on literature-derived knowledge on the direction of regulation (either activating or inhibiting).

## RESULTS

### Transcriptional responses after a control session

First we assessed the homogeneity of transcriptional responses in the control session. For this, we analyzed whether the transcriptional responses of participants belonging to edu classes 3–4 ($n = 4$) and edu classes 1–2 ($n = 6$) differed in the control session. Only 4 genes were found to be differentially expressed, suggesting no major differences in the transcriptional responses between groups to a 'non-musical activity.' Therefore, we used the expression data of the whole 'music-free' control group as a reference for the comparative analyses.

We used Spearman's rank-based correlation to check for each gene, whether the transcriptional responses correlated with the age or sex of the participants in either the listening session ($n = 48$) or the control sessions ($n = 10$). After multiple testing corrections, we found no significant effects of age or sex on the transcriptional responses.

### Transcriptional response after listening to music

Based on neuroscientific studies, the brains of musicians and non-musicians differ structurally and functionally (*Elbert et al., 1995*; *Gaser & Schlaug, 2003*). This led us to ask whether the transcriptional responses of musically experienced participants would differ from those of musically inexperienced participants when listening to music. Therefore, we compared the transcriptional responses of listening to music separately for musically experienced and inexperienced participants vs controls. Comparisons were made based on two subphenotypes of musical experience: musical aptitude and music education.

First, we compared the magnitude of pre-post fold-changes in the genome-wide transcriptional profiles of *listeners of edu classes 3–4* ($n = 29$) and *controls* ($n = 10$). Using RankProd non-parametric statistics and stringent selection criteria, we identified 45 differentially expressed genes (27 up-regulated and 18 down-regulated). Next, we compared the genome-wide transcriptional profiles of *listeners with high COMB scores* ($n = 12$) and *controls* ($n = 10$). Similar statistical analysis identified 97 differentially expressed genes (75 up-regulated and 22 down-regulated). The differentially expressed genes from both the comparisons are listed in Table S2, and a comparison of the pre-post changes in both conditions is shown in Fig. 1.

#### Functional annotation

Based on gene ontology analyses (Table S3), the genes up-regulated in the *listeners of edu classes 3–4* are known to be primarily involved in the regulation, secretion and

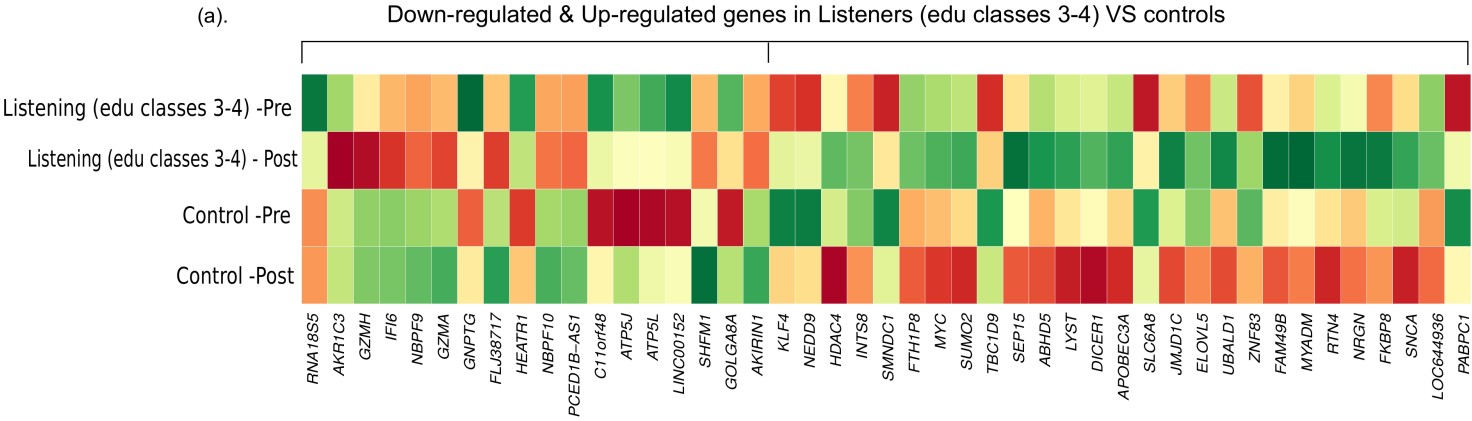

(a). Down-regulated & Up-regulated genes in Listeners (edu classes 3-4) VS controls

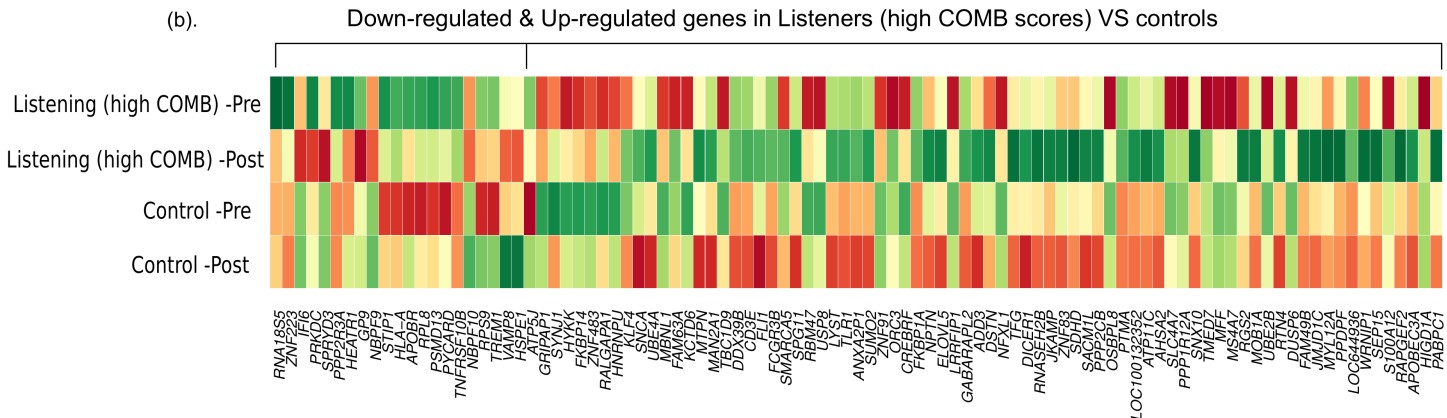

(b). Down-regulated & Up-regulated genes in Listeners (high COMB scores) VS controls

**Figure 1** **Differential gene expression in experienced listeners vs 'music-free' controls.** Heatplot representations of mean expression values pre- and post-music listening session and control sessions. The red-yellow-green palette represents low-moderate-high expression values. (A) Educated listeners vs controls, (B) Competent listeners vs controls.

transport of the neurotransmitter dopamine (e.g., *SNCA*, *RTN4*, and *SLC6A8)*, protein sumoylation (*SUMO2* and *HDAC4*) and neuron projection (*SNCA*, *RTN4*, *DICER1* and *MYC*). Down-regulated genes are known to affect functions such as mitochondrial ATP synthase-coupled proton transport and cytolysis (e.g., *ATP5J, ATP5L, GZMH*, and *GZMA*). Several of the genes, including the dopamine secretion-related genes (*SNCA*, *RTN4*) up-regulated in *listeners of edu classes 3–4*, were also found to be up-regulated in *listeners with high COMB scores*. Here, we should note that the *COMB scores* are strongly correlated with music *edu classes* (Spearman's rho 0.5644; *p*-value 2.931e–05). In *listeners with high COMB scores*, gene ontology classification revealed that the up-regulated genes are involved in functions such as long-term synaptic potentiation (*NPTN* and *SNCA*), dephosphorylation and regulation of cell communication. Down-regulated genes are known to be involved in functions such as positive regulation of caspase, peptidase and endopeptidase activities (Table S3).

We further performed Entrez gene annotation and an extensive literature survey for all the genes that are differentially expressed after listening to music (in listeners of both edu

**Table 2  Putative biological functions of the differentially expressed genes after listening to music.**

| Biological function | Gene | Direction of regulation |
|---|---|---|
| Dopamine secretion, transport, signaling | *SNCA, RTN4, RGS2, SLC6A8* | Up |
| Synaptic neurotransmission (Vesicular exocytosis, endocytosis) | *SNCA, STXBP2, FKBP8, SYNJ1, LYST, SUMO2, HDAC4, DUSP6* | Up |
| Synaptic function | *SNCA, NPTN, FKBP8, NRGN, HDAC4* | Up |
| Learning and memory, cognitive performance | *SNCA, NRGN, NPTN, FKBP8, RTN4, SLC6A8, NEDD9* | Up |
| Song learning and singing in songbirds | *SNCA, NRGN, RGS2, MYC, UBE2B* | Up |
| Auditory cortical activation | *HDAC4, LRRFIP1* | Up |
| Absolute pitch | *FAM49B, HDAC4* | Up |
| Neuroprotection | *SNCA, RTN4, FKBP8, SLC6A8, KLF4* | Up |
| Neurogenesis | *KLF4, SMNDC1, S100A12* | Up |
| Neuronal apoptosis | *CASP8, GZMH, GZMA, IFI6, PYCARD, TNFRSF10B, HSPE1* | Down |
| ATP synthase coupled proton transport | *ATP5J, ATP5L* | Down |

classes 3–4 and high COMB scores). This revealed that the up-regulated genes are known to be associated with dopamine signaling, synaptic neurotransmission, synaptic function, learning, memory and cognitive performance, song learning and singing in songbirds, auditory cortical activation, absolute pitch, neuroprotection and neurogenesis (Tables 2 and S4). On the other hand, down-regulated genes are known to cause mammalian neuronal apoptosis, immoderate oxidative phoshorylation and deficits in dopaminergic neurotransmission, which are the characteristics of neurodegeneration (Tables 2 and S4). Detailed information about the putative biological functions of the differentially expressed genes is provided in Table S4.

### Upstream regulator analysis

To obtain insight into the molecules that might mediate the observed differences in gene expression, we performed an upstream transcription regulator analysis using IPA (Table S5). This analysis revealed that the up-regulated genes in *listeners of edu classes 3–4* significantly overlap the known target genes of the glucocorticoid receptor *NR3C1* ($p$-value 0.001), and progesterone receptor *PGR* ($p$-value 0.0008), whereas the down-regulated genes did not display any significant overlap. In *listeners with high COMB scores*, upstream regulator analysis again identified the up-regulation of target genes of the glucocorticoid receptor gene (*NR3C1*) and also the target genes of several other transcription regulators such as *TP53, MYC, HOXA9, CD24* and chorionic gonadotropins. Down-regulated genes significantly overlapped the known target genes of pro-inflammatory transcription regulators such as tumor necrosis factor (*TNF*), a member of its superfamily (*TNFSF10*), interferon gamma (*IFNG*), a member of its gene cluster (*IFNA2*), the microtubule-associated protein tau (*MAPT*)and the nuclear factor kappa B family gene (*NFKB1A*; Table S5).

### Transcriptional responses of participants with no significant experience

Furthermore, we repeated similar analyses to compare the magnitude of pre-post fold-changes over time in *listeners of edu classes 1–2 vs controls* and *listeners with low COMB scores vs controls*. Using the same analysis methods and selection criteria to identify the differentially expressed genes, we identified 8 and 22 differentially expressed genes, respectively in the comparisons. However, functional characterization of those genes did not reveal any significant findings.

## DISCUSSION

The findings of this study suggest that listening to classical music has an effect on human transcriptome. The up-regulation of genes related to dopamine secretion and signaling is in agreement with the previous neuroimaging-based evidences (*Salimpoor et al., 2011*). Particularly, *alpha-synuclein* ($\alpha$-*synuclein*; *SNCA*), one of the most up-regulated genes, is involved in dopamine (DA) neuronal homeostasis (*Murphy et al., 2000*; *Oczkowska et al., 2013*). Interestingly, *SNCA* is located on chromosome 4q22.1, the most significant region of linkage for musical aptitude (*Pulli et al., 2008*; *Oikkonen et al., 2014*) and regulated by *GATA2* (*Scherzer et al., 2008*), which is associated with musical aptitude (*Oikkonen et al., 2014*) (Fig. 2). These data provide convergent evidence about the molecular basis of musical traits from both DNA and RNA studies. Another finding from the upstream regulator analysis suggests that listening to music primarily increased the expression of the target genes of the glucocorticoid receptor (*NR3C1*). Notably, dopaminoceptive neuronal glucocorticoid receptor has been described as a key molecule in the regulation of addictive behavior. By reducing dopamine re-uptake, *NR3C1* increases the synaptic concentration of dopamine, which leads to rewarding and reinforcing properties (*Ambroggi et al., 2009*) that have previously been linked to listening to music (*Blood & Zatorre, 2001*; *Koelsch, 2011*; *Koelsch, 2014*; *Salimpoor et al., 2011*; *Salimpoor et al., 2013*).

The up-regulation of genes related to synaptic vesicular exocytosis, endocytosis, neurotransmission and plasticity seems perfectly rational here because the majority of these biological processes are essential for the secretion and signaling of neurotransmitters (*Südhof & Rizo, 2011*; *Saheki & De Camilli, 2012*). As listening to music has been known to induce the secretion and signaling of a neurotransmitter, dopamine (*Sutoo & Akiyama, 2004*; *Salimpoor et al., 2011*), we can speculate the role of these up-regulated genes in facilitating dopaminergic neurotransmission after listening to music. Moreover, some of the up-regulated genes have evident roles in enhancing cognitive functions like long-term potentiation (LTP) and memory. In previous behavioral studies, music education and training have proven to have beneficial effects on cognitive development, cognitive performance, verbal and long-term memories (*Rammsayer & Brandler, 2003*; *Schlaug et al., 2005*; *Sluming et al., 2007*; *Wong et al., 2007*; *Roden, Kreutz & Bongard, 2012*; *Rodrigues, Loureiro & Caramelli, 2013*).

Several of the differentially expressed genes have been demonstrated to be responsible for song learning and singing in songbirds (*Wada et al., 2006*), which suggests a possible evolutionary conservation in biological processes related to sound perception. Further-

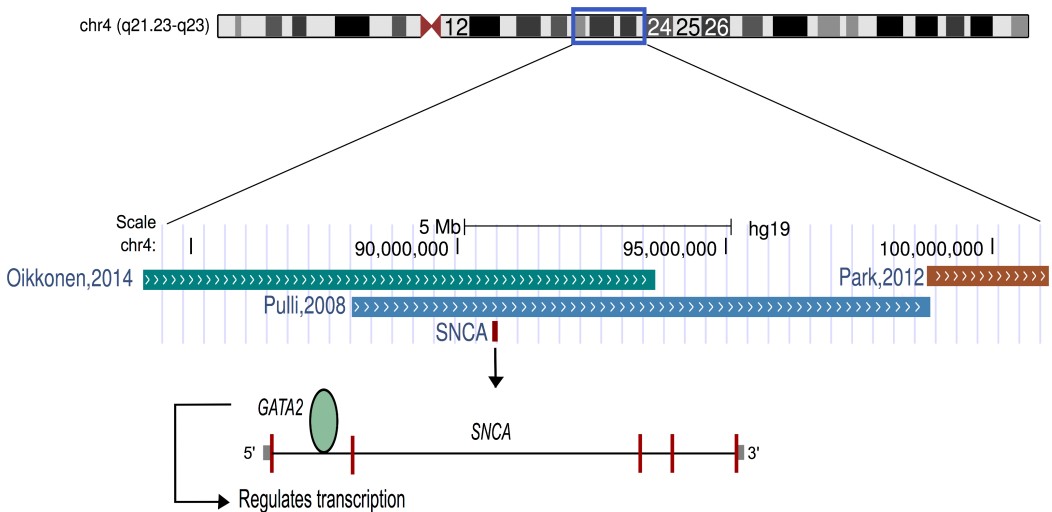

**Figure 2 Schematic representation of chromosome 4.** The $\alpha$-synuclein gene (*SNCA*) that was found to be up-regulated after music perception in this study is located in the best linkage region of musical aptitude as shown by *Pulli et al. (2008)*, *Park et al. (2012)* and *Oikkonen et al. (2014)*. *GATA2*, which is located in the best genome-wide association region of musical aptitude (*Oikkonen et al., 2014*) and regulates the *SNCA*, is also shown.

more, the up-regulation of genes associated with human auditory cortical activation (*Renvall et al., 2012*) and absolute pitch (*Theusch, Basu & Gitschier, 2009*; *Gervain et al., 2013*) are logical, because listening to music involves both of those auditory phenomena.

Auditory perception processes have been known to exhibit convergent evolution across species. Notably, the human auditory center is identical to those of the first primates who inhabited the planet millions of years ago (*Langner , 1992*; *Montealegre-Z et al., 2012*). In addition, widespread adaptive convergent sequence evolution has been found recently in hearing-related genes in echolocating bats and dolphins (*Parker et al., 2013*). Similarly, convergent sequence evolution has also been identified in vocal-learning birds and mammals (*Zhang et al., 2014*). More recently, convergent gene expression specializations have been detected in songbirds and humans in the regions of brain that are essential for auditory perception and speech production (*Pfenning et al., 2014*). Thus, the genes detected by *Pfenning et al. (2014)*, in general, represent the genes belonging to auditory perception pathway in both songbirds and humans. Here, genes belonging to the auditory perception pathway ($\sim$2-fold enrichment; *p*-value: 0.028; two-sided Fisher's exact test) were found to be enriched among the genes that are differentially expressed after listening to music. This suggests that our results serve as a relevant molecular background for music perception in humans.

The widely-documented neuroprotective role of some of the up-regulated genes and the down-regulation of several neurodegeneration-inducing genes support the notion of a neuroprotective role for music and may provide a working mechanism for the use of music therapy, especially in treating neurodegenerative diseases (*AMTA, 2014*; *Conrad, 2010*; *Holmes, 2012*).

In this study, significant transcriptional responses were observed only in individuals who either have substantial periods of music education/training or have relatively higher musical aptitude scores. This suggests that certain musical abilities (either innate or acquired through music education) may influence the transcriptional responses of listening to music. Previous works have shown that the familiarity of music attained through music education or repeated music exposure is known to largely influence the rewarding aspects of listening to music (*Sarkamo et al., 2008*; *Salimpoor et al., 2009*; *Van den Bosch, Salimpoor & Zatorre, 2013*; *Schubert, Hargreaves & North, 2014*). Here, we acknowledge that the effect of music exposure on human gene expression could be very subjective and may vary depending on several factors such as age, sex, culture, previous listening habits, music education and training and personal liking of music, as recently discussed (*Wong et al., 2007*; *Salimpoor et al., 2009*; *Van den Bosch, Salimpoor & Zatorre, 2013*; *Mikutta et al., 2014*). To be able to comprehensively characterize the transcriptomic alterations of music-listening, further studies are required to assess the effect of listening to different genres of music, at different ages, in different ethnicities and in individuals with varying music education levels and listening habits, with varying durations of listening.

The participants, who listened to music in this study, were unaware of the type of music that was intended for the listening session. Similarly, without further details about the intended activity, the participants were invited to attend a 'music-free' control session. Here, it is crucial to discuss the association between 'anticipation' and 'listening to music.' Anticipation is always involved in listening to music (*Salimpoor et al., 2011*). The unveiling of series of tones with time evokes anticipatory responses because of the cognitive expectations and prediction cues (*Huron, 2006*; *Meyer, 2008*). Functional neuroimaging studies have successfully distinguished the anatomical regions that respond to anticipation and consumption of music (*Salimpoor et al., 2011*). However, distinguishing the human gene expression signatures of anticipatory and consummatory responses of music is not yet feasible. Therefore, even if we perform a blinded experiment here, we might not be able to exclude the effect of anticipation or expectation.

As brain samples are inacesssibile in this type of study, we used peripheral blood as a window to the study the effects of listening to music. Peripheral blood is known to share more than 80% of the transciptome and significant gene expression similarities with other tissues including multiple regions of brain (*Liew et al., 2006*; *Sullivan, Fan & Perou, 2006*; *Tylee, Kawaguchi & Glatt, 2013*). Thus, peripheral blood could certainly provide surrogate information concerning gene expression in brain tissue for a subset of genes (*Davies et al., 2009*). For instance, the molecular alterations in dopamine metabolism and mitochondrial function, which are the potential hallmarks of Parkinson's disease, have been detected in peripheral blood (*Scherzer et al., 2007*). Notably, genes that are responsive to physiological stimuli (which are earlier thought to be tissue-specific) and genes involved in neuroendocrine pathways (e.g., hormone receptors, neurotransmitter receptors) are expressed in the peripheral blood. Because of these characteristics, peripheral blood has been used as a proxy in several studies when a specific tissue is not available (e.g., human brain), especially in behavioral and neurodegenerative studies (*Mohr & Liew, 2007*). In

the wake of all these findings, a subset of the molecular mechanisms identified here may legitimately reflect the transcriptomic alterations in brain after listening to music.

## ACKNOWLEDGEMENTS

We thank all of the participants for their generous cooperation. We thank Päivi Onkamo for her constructive comments about the manuscript. We are grateful to Petri Myllynen, Sanna Pyy, Laura Salmela, Sonja Suhonen, Jaana Oikkonen and Kai Karma for expert technical help. We thank the High-Throughput Genomics Group at the Wellcome Trust Centre for Human Genetics for the generation of the Gene Expression data.

### Funding

The Academy of Finland (grant reference #13371) and the Biocentrum Helsinki Foundation supported this work. The funders had no role in study design, data collection and analysis, decision to publish, or preparation of the manuscript.

### Grant Disclosures

The following grant information was disclosed by the authors:
The Academy of Finland: #13371.
Biocentrum Helsinki Foundation.

### Competing Interests

The authors declare there are no competing interests.

### Author Contributions

- Chakravarthi Kanduri analyzed the data, wrote the paper, prepared figures and/or tables, reviewed drafts of the paper.
- Pirre Raijas and Liisa Ukkola-Vuoti contributed reagents/materials/analysis tools, reviewed drafts of the paper.
- Minna Ahvenainen and Anju K. Philips performed the experiments, reviewed drafts of the paper.
- Harri Lähdesmäki conceived and designed the experiments, reviewed drafts of the paper.
- Irma Järvelä conceived and designed the experiments, wrote the paper, reviewed drafts of the paper.

### Human Ethics

The following information was supplied relating to ethical approvals (i.e., approving body and any reference numbers):

The Ethics Committee of Helsinki University Central Hospital approved this study (permission number 233/13/03/2013). Written informed consent was obtained from all the participants.

## Microarray Data Deposition

The following information was supplied regarding the deposition of microarray data:

The data reported in this paper have been deposited in the Gene Expression Omnibus (GEO) database, www.ncbi.nlm.nih.gov/geo (accession no. GSE48624).

## Supplemental Information

Supplemental information for this article can be found online at http://dx.doi.org/10.7717/peerj.830#supplemental-information.

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
