# Peer review of "The effect of listening to music on human transcriptome"

_PeerJ, doi:10.7717/peerj.830_

## Round 0.1 · original submission · Major Revisions

· Academic Editor

Major Revisions

I have now received two thorough reviews of your paper. Both think the paper well-written and very interesting. I agree with this sentiment. Both, however, have some major concerns that must be addressed before your paper can be considered further. Some of these suggestions will require additional lab work and more detail on some aspects of the study to justify the conclusions.

·

Basic reporting

Overall this aspect of the report is acceptable to excellent. Here are a couple concerns for improving communication:
1. uniform use of terminology to refer to study subgroups, e.g., 0-3 on Supplemental table 1 vs. edu classes 1-4 in methods; and educated listeners in figure. Also in results specify that italicized "listeners..." is the group that listened to the music. It was not quite as obvious as the authors assumed.
2. Temporal aspect of response to music - blood samples were taken at 20 minutes (or within ? minutes) upon completion of listening to a single music composition excerpt. Please reference other studies, e.g., meal challenge http://ajcn.nutrition.org/content/91/1/208.full, and temporal sample time (6 hours post challenge in this case). I am assuming non-scientific persons will be readers of this article and they need to be informed that 20 minutes is a really short time period for a sitmulus to get from the ear to the genes in blood cells to change mRNA levels significantly. Seems impossible to this scientist.

Experimental design

-Were subjects blind to treatment before the music started? The anticipation of listening to music might otherwise be the bias on the effect observed.
-Were the same instructions given to the group called in for listening as stated in the methods section for the control group?
-Some control subjects were included in both the listening-vs-control study, i.e., six edu-class 4-3 (2-3 on suppl table 1) were in both groups. You should state specificially which 5 of the 15 control group subjects were dropped out.

Validity of the findings

I have two major concerns:
1. 20 minutes from ear to gene in blood cell nuclei is extremely difficult to believe; what positive control could be used? ... extreme fear? ...the opposite of calming music tested in this study.

2. 12 of the 15 control subjects are 49 years or older. 10 of 29 edu-class 3-4 were older than the average age of the entire study group. 4 of the 12 highest COMB group were older than the average age... I am not confident that the potential effect of age has been evaluated correctly for the comparisons made resulting in statistically significant differences in differential gene expression.

3. The subjects must have been blinded to the intended activity when called in for both the control and listening session. If not the data are invalid - a bias can not be ruled out.

·

Basic reporting

Kanduri et al are presenting a quite fascinating hypothesis that music exposure can affect the RNA content in peripheral blood. The paper is clearly written and the topic is of interest to a wide audience. However, I feel the authors should have a more solid support for their rather controversial claim.

Experimental design

My main concern is that the up- and down-regulated genes after music exposure have not been experimentally validated by alternative methods. I would like to see a validation by a more exact method like qrt-PCR for some of the differentially expressed genes reported in the manuscript. This validation should be done on all samples, i.e. both in music exposure and control studies, to show that expression changes are consistently seen in participants exposed to music while no changes are observed in the control study. Also, the expression differences between musically experienced and inexperienced participants should be validated in the same way. Without such validations, all findings are based purely on statistical analyses of gene expression microarray data. In my experience such data can contain a lot of noise due to experimental factors that are difficult to control. This could potentially be a problem for the statistical analyses, especially when studying rather modest fold-changes in gene expression as is the case in this study.

Validity of the findings

The sentence suggesting that HDAC and GZMA are under positive selection in individuals with high COMB scores (lines 208-211) should be removed or supported by a reference to published work.

Regarding the need for validation of differential expression by alternative methods, see comments above.

Comments for the author

One very important limitation of this study is that peripheral blood was used for measuring the transcriptional changes that are believed to mainly take place in the brain. The authors have commented on this and included a couple of references supporting the claim blood can act as a surrogate for measuring RNA expression in the brain. However, since this is a fundamental question for this whole study it would be good to know whether there exist any recent studies in the literature (ideally performed using high-throughput RNA sequencing) where the correlation between brain-blood gene expression has been examined? If that is the case, the results of those studies should be referenced either in the Introduction or Discussion. The recent studies by Aberg et al and Ewald et al, mentioned towards the end of the Discussion, seems to have examined DNA methylation and not gene expression.

---

## Round 0.2 · accepted · Accept

· Academic Editor

Accept

Thank you for your careful and functional attention to the previous comments by reviewers. I feel these comments and your adjustments have improved the paper for general readership. I also appreciate you addressing some of the comments with very recent literature additions to your paper. Very nice. I think your arguments are well supported. Well done.

The article is accepted, subject to you making one final clarification to your text. I would like you to make it clear in the final text how your work addressed the concerns of Dr Felix regarding the blinding of the subjects. You adequately dealt with this question in your rebuttal letter (but not in the main text) and so I believe that the easiest thing to do here is to simply copy the following paragraph from your rebuttal letter into the the discussion section of the paper:

"The participants who listened to music were unaware of the type of music that was intended for the listening session (as already mentioned in the methods section of the manuscript). Similarly, without further details about the intended activity, the participants were invited to attend a ‘music-free’ control session. Here, it is crucial to discuss the association between ‘anticipation’ and ‘listening to music’. Anticipation is always involved in listening to music (Salimpoor et al., 2011). The unveiling of series of tones with time evokes anticipatory responses because of the cognitive expectations and prediction cues (Huron, 2006; Meyer, 2008). Functional neuroimaging studies have successfully distinguished the anatomical regions that respond to anticipation and consumption of music (Salimpoor et al., 2011). However, distinguishing the human gene expression signatures of anticipatory and consummatory responses of music is not yet feasible. Therefore, even if we perform a blinded experiment here, we might not be able to exclude the effect of anticipation or expectation."

Note: You have references within this paragraph, so that might mean reworking the reference list too.